# Public Health Workforce Gaps, Impacts, and Improvement Strategies from COVID-19

**DOI:** 10.3390/ijerph192013084

**Published:** 2022-10-12

**Authors:** Chelsey Kirkland, Kari Oldfield-Tabbert, Harshada Karnik, Jason Orr, Skky Martin, Jonathon P. Leider

**Affiliations:** 1Center for Public Health Systems, Division of Health Policy and Management, School of Public Health, University of Minnesota, Minneapolis, MN 55455, USA; 2Local Public Health Association of MN, Saint Paul, MN 55103, USA

**Keywords:** COVID-19, public health workforce, focus groups, mixed-methods, gaps, impacts, improvement strategies

## Abstract

The public health workforce has been instrumental in protecting residents against population health threats. The COVID-19 pandemic has highlighted the importance of the public health workforce and exposed gaps in the workforce. Public health practitioners nationwide are still coming to understand these gaps, impacts, and lessons learned from the pandemic. This study aimed to explore Minnesota’s local public health practitioners’ perceptions of public health workforce gaps, the impacts of these workforce gaps, and the lessons learned in light of the COVID-19 pandemic. We conducted seven concurrent focus groups with members of the Local Public Health Association of Minnesota (LPHA; *n* = 55) using a semi-structured focus group guide and a survey of the local agencies (*n* = 70/72 respondents, 97% response rate). Focus group recordings were transcribed verbatim and analyzed using deductive and inductive coding (in vivo coding, descriptive coding), followed by thematic analysis. The quantitative data were analyzed using descriptive analyses and were integrated with the qualitative data. Participants indicated experiencing many workforce gaps, workforce gaps impacts, and described improvement strategies. Overall, many of the workforce gaps and impacts resulting from COVID-19 discussed by practitioners in Minnesota are observed in other areas across the nation, making the findings relevant to public health workforce nationally.

## 1. Introduction

The public’s vanguard defense against threats to population health, from disease and social service safety nets to environmental hazards, is the governmental public health workforce. The unique responsibilities of governmental public health are outlined by the Foundational Public Health Services [1]. A key responsibility of local public health is emergency preparedness, and response. Thus, during COVID-19 these entities assumed most of the responsibility for testing, contract tracing, community communication, vaccination, and overall coordination [2]. Addressing COVID-19 and population health concerns through collective services requires skilled professionals across different levels of government. However, the national public health enterprise has struggled for decades to recruit and retain skilled professionals due to myriad challenges—competition from the private sector, visibility of governmental public health, and staff burnout, to name a few. At each governmental level, the public health workforce has atrophied following years of underinvestment, disinvestment, and separations [3,4,5], which the COVID-19 pandemic has exacerbated, increasing negative public sentiment toward public health and increasing the prevalence of bullying and harassment of its workers [6,7,8].

Currently, the public health workforce is experiencing turmoil as many workers express various dissatisfactions and are planning to leave their jobs, leaving many vacant positions [9]. In spring 2021, the Centers for Disease Control and Prevention (CDC) conducted a nationally representative survey of the governmental public health workforce, which found that more than half of respondents reported symptoms of at least one adverse mental health condition, many of whom were unable to take time off or worked more than a standard 40-hour work week [10]. A later survey of workers between September 2021 and January 2022 similarly found more than half of respondents reported having at least one post-traumatic stress disorder symptom, and nearly one-third were considering leaving their health department within the next year [11]; among other substantial mental health challenges [12]. Compared to other public sectors, governmental public health faces the unique challenge of responding to the pandemic with a decreased workforce and insufficient resources [13]. A consequence of this has been a further weakening of the public health system through retirements and other departures [11].

Minnesota exemplifies these trends and consequences with the local public health workforce facing substantial challenges [14,15]. Minnesota’s public health workforce gaps have weakened the state’s public health system, and public health workforce capacity varies geographically [16]. Public health workforce gaps are found throughout the state and heightened in rural areas where resources are more scarce and local agencies have difficulties competing for skilled professionals [16]. In 2019, a Minnesota Department of Health (MDH) [17] survey of local public health and public health scholars found that the top needed positions included program-specific positions, directors, and assessment and planning. To reduce workforce gaps, they recommended revamping student practicum placement processes, filling agency director vacancies, increasing the workforce’s education, and improving systemic workforce changes. Through another survey conducted in August–October 2020 by the Local Public Health Association of Minnesota (LPHA), respondents were asked to rate their local health department’s staffing capacity in preparing for and responding to COVID-19. Of those responding (*n* = 66), most rated their health department’s capacity as ineffective (33%) or neutral (36%) [18].

Following the onset of the COVID-19 pandemic, communities emphasized rethinking the social services safety net and creating lasting and adaptable change to the new economic, social, and public health challenges [19]. Thus, an investigation is needed to determine which occupations are of highest priority and which workforce interventions are most needed to fill those gaps, particularly to address the consequences of increasing unemployment and economic disruption due to the COVID-19 pandemic. Further, with a dynamic workforce, there is a growing need for establishing a strong pipeline connecting skilled professionals to public health agencies such as identifying which public health occupations are in the highest need and what barriers exist to ensuring adequate staffing of local public health. In light of this, the present study aimed to explore local public health practitioners’ perceptions of public health workforce gaps, the impacts of workforce gaps, and lessons learned from the COVID-19 pandemic.

## 2. Materials and Methods

The study design is classified as an “Exploratory” mixed-methods, (QUAL -> quant-> integration model), wherein qualitative data were collected before quantitative data and then the qualitative and quantitative data results were integrated [20]. We collected the qualitative data through focus groups, quantitative data through a cross-sectional survey, and integrated those data for the mixed-methods analysis.

### 2.1. Qualitative Data

We conducted seven virtual, concurrent, semi-structured focus groups with local health department (LHD) staff during an LPHA general membership meeting in January 2022. All LPHA members received information about the opportunity to participate in the focus groups prior to the meeting and all members present during the meeting were invited to participate. Participants were largely directors, administrators, managers, or supervisors in their LHDs. Focus groups were chosen to gain insights into participants’ shared understanding and perceptions [21] of the public health workforce. Focus group participants were representatives of city and county public health agencies across the state. Focus group participants received information as part of the general meeting agenda, which included study information and an invitation to participate in the focus groups. Focus group participants received an e-mail reminder one day prior with an information sheet and video conference meeting information.

All focus group participants logged into the main virtual meeting room where the lead researcher gave an overview of the project, introduced the researchers conducting the project, and answered questions about the study. Focus group participants were then randomly assigned to one of the seven breakout rooms. Trained moderators facilitated each breakout room, reminded focus group participants about the purpose of the focus groups, answered any questions, and obtained verbal consent to record the focus group. Moderators followed a semi-structured focus group guide developed using literature around public health workforce gaps with input from LPHA leadership and public health workforce experts. Focus groups lasted approximately one hour, with 5–10 focus group participants each, after which focus group participants returned to the main video conferencing room for a large group debrief session.

#### 2.1.1. Moderator Training

Following best practices [22], moderators received training before the focus groups to standardize data collection protocols. During the training, the research team provided moderators with an overview of the project, the focus groups’ purpose, the discussion guide, focus group logistics, technology in use, and contingency plans. The research team and moderators participated in a mock focus group session before the event.

#### 2.1.2. Analysis

Moderators recorded focus groups using the video conferencing software and saved all chat files. Following the event, researchers transcribed audio files verbatim and uploaded transcripts and chat files into Dedoose (SocioCultural Research Consultants, LLC, Los Angeles, CA, USA) for analysis. Data were analyzed using deductive and inductive approaches [23]. We then created three predetermined (“deductive”) codes as domains based on the study purpose: workforce gaps, impacts of workforce gaps, and improvement strategies. Codes were developed throughout the analysis (“inductive”) within each domain and consisted of in vivo coding [24], descriptive coding [24], and thematic analyses [25]. Domains and themes cut across all focus group participants and are numbered in ascending order in relation to the domain (e.g., the two themes under domain 1 are numbered 1.1 and 1.2).

### 2.2. Quantitative Data

A cross-sectional survey was developed based on the focus group results from the first two themes to quantify and further explore public health workforce gaps and impacts of the COVID-19 pandemic. We conducted cognitive interviews with three local public health workforce experts and former local health officials to ensure survey respondents would comprehend the questions and question choices were consistent with the researchers’ aims. The survey was pre-tested by three retired public health officials. We then e-mailed a link to the final survey to LPHA member agencies (*n* = 72), which was administered June–July 2022 using Qualtrics (Provo, UT, USA). The three survey questions analyzed for this research were developed using previous public health workforce gaps literature:“Before the COVID-19 pandemic (i.e., before March 2020), if and when adequate funding was available to increase staff capacity, were you concerned about any of the following items?” (survey respondents were given 17 different choices and were instructed to select all that apply);“Which of the following concerns increased during or after the pandemic, i.e., after March 2020?” (survey respondents were presented with the same list of choices as the previous question and were instructed to select all that apply); and“After the COVID-19 pandemic started (i.e., after March 2020), did staffing shortages at your agency lead to any of the following?” (survey respondents were provided 14 different choices and again instructed to select all that apply).

Survey data were downloaded from Qualtrics and used Stata Version 17.1 (College Station, TX, USA) to analyze the data using descriptive statistics.

### 2.3. Data Integration

After the qualitative themes were developed and quantitative data descriptive statistics were completed, the qualitative and quantitative data were integrated by sorting the survey response choices from the survey questions by focus group themes. The first two survey questions were combined into domain 1, and survey question responses were then sorted into themes 1.1 and 1.2. The third survey question responses were sorted into themes 2.1, 2.2, and “other”. Thus, in domain 1 (“workforce gaps”), the survey question response choices about staffing concerns before COVID-19 and whether those concerns increased during or after COVID-19 were sorted by whether they related to a “local infrastructure gap” (theme 1.1) or “workforce capability and capacity gap” (theme 1.2). For domain 2 (“workforce gaps impacts”), the response choices for staffing shortage impacts were sorted by whether they related to a “poor operational outcome from gaps” (theme 2.1), “adverse personnel impact” (theme 2.2), or “other” categorization. No survey questions aligned with domain 3.

The study was classified as not human subject research by the researchers’ Institutional review board (IRB).

## 3. Results

A total of 55 LHD staff participated in the seven concurrent focus groups. In the survey portion, 70 LHDs completed the survey (97% response rate).

Through qualitative analysis, the researchers developed seven themes (local infrastructure gaps, workforce capability and capacity gaps, poor operational outcomes due to workforce gaps, adverse personnel impacts, retention strategies, systems-level changes, and recruitment/hiring strategies) that were concordant with the three domains (Table 1). Each theme was developed within the context of the COVID-19 pandemic and described the impacts of the pandemic, including challenges, lessons learned, and opportunities for the public health workforce.

### 3.1. Domain 1: Workforce Gaps

The first domain, workforce gaps, was defined as challenges health agencies were experiencing among their workforces due to the COVID-19 pandemic. Two themes from the focus groups were developed: local infrastructure gaps (theme 1.1) and workforce capability and capacity gaps (theme 1.2). Theme 1.1, local infrastructure gaps, was defined as structural or procedural deficiencies that focus group participants identified within their local public health agencies. Some of these needs included inefficient or ineffective local processes and constraints limiting competitiveness. For example, one focus group participant described the challenge of local health department position benefits not being attractive to candidates anymore since it does not outweigh the challenges COVID-19 has brought to the department saying, “…the benefits. They’re not that much better than what they can get somewhere else, right? So we’re touting of all [benefit packages such as pensions], yes, but we have great benefits. That just doesn’t hold water for people anymore [after COVID-19]”.

The second theme, workforce capability and capacity gaps, was defined as needing professionals with appropriate skills (“capability”) and having enough staff to deliver services (“capacity”). These capability and capacity gaps were recurrent and may have worsened during the COVID-19 pandemic due to increased turnover. One focus group participant provided an example of this,


*[Name] did it before. I don’t know how to do it and she didn’t have time to teach people before she left. So we’ve had to kind of cut out the ability to share a lot of data with the county.*


Survey participants’ identified workforce gaps were similar to those identified by focus group participants (Figure 1). The largest workforce gap indicated by survey respondents, which was present before and increased during or after COVID-19, was experiencing a small number of applicants (*n* = 53 before [COVID-19] and *n* = 54 increased [during or after COVID-19]). The same number of survey respondents who indicated a concern before COVID-19 often indicated the concern increased during or after COVID-19, though this observation did not hold true for each staffing capacity concern. For example, among participants who indicated receiving authorizations for new positions as a gap before COVID-19, the second most common (*n* = 48), only 34% of survey respondents (*n* = 24) indicated an increase in this gap during or after COVID-19. The workforce gap that increased the most during or after COVID-19 was the staffing capacity to onboard new employees (*n* = 27, *n* = 38 before and increased, respectively). Lastly, the range for the number of gaps survey respondents indicated their agencies experienced was between 0 (*n* = 1) and 14 (*n* = 1) with most survey respondents indicating experiencing 5 (*n* = 11) different gaps. For number of gaps that increased during or after COVID-19, the range was also between 0 (*n* = 2) and 14 (*n* = 1). Most participants indicated that 2 (*n* = 10) gaps increased.

When integrating the data, 94% of survey respondents (*n* = 65) indicated that local infrastructure gaps (focus group theme 1.1) and 93% of survey respondents (*n* = 64) indicated that workforce capability and capacity gaps (theme 1.2) were present before COVID-19. Furthermore, 81% of survey respondents (*n* = 55) indicated that local infrastructure gaps and 94% of participants (*n* = 64) indicated that workforce capability and capacity gaps increased during or after COVID-19. Within these two themes, the top workforce gap within local infrastructure gaps was in receiving authorizations for new positions (*n* = 48, 70%) and 35% (*n* = 24) indicated that this gap increased during or after COVID-19. The top workforce capability and capacity gap was a small number of applicants (*n* = 53, 77%) and 79% (*n* = 54) indicated that this gap increased during or after COVID-19. Please see Table 2 for the associated statistics-by-theme joint table.

### 3.2. Domain 2: Workforce Gaps Impacts

The second domain, workforce gaps impacts, captured the consequences of the gaps in the workforce. There were two themes developed for the focus groups encapsulating these workforce gaps impacts: poor operational outcomes due to workforce gaps (theme 2.1), and adverse personnel impacts (theme 2.2). Theme 2.1, poor operational outcomes due to workforce gaps, was defined as consequences to focus group participants’ agency operations due to unfilled staff positions. Most focus group participants described their agencies as experiencing poor operational outcomes. For example, one focus group participant explained that the public versus private wage gap created by the COVID-19 pandemic caused their agency to lose an entire program, “we actually had to give our program back to the state because we cannot find a [position]. And largely we did have applicants, but we couldn’t pay them what they wanted to be paid”.

Almost every focus group participant described agency-level adverse personnel impacts resulting from workforce gaps, which led to the development of theme 2.2, adverse personnel impacts, defined as the negative effects related to staff experienced by focus group participants’ agencies due to unfilled staff positions. Reduced satisfaction and retention was one of the underlying causes of many other workforce gaps, which worsened employee satisfaction and retention even more. For example, one focus group participant spoke about retention challenges by saying, “where’s our hazard pay, hey where’s our overtime pay? A lot of us [agencies] aren’t able to offer things like that, which is also very attractive in retaining [employees]”.

The overwhelming majority of survey respondents (90%) indicated that the primary impact of workforce gaps was staff assuming additional responsibilities (*n* = 64). After that, over three-quarters of survey respondents indicated that workforce gaps caused staff to burnout (*n* = 59, 83%), and agencies scaled back their programs and services (*n* = 54, 76%). Despite all of the negative impacts agencies experienced due to workforce gaps, only one experienced a suspension or withdrawal of funds by donors. Furthermore, 3% (*n* = 2) indicated that their agencies did not experience any workforce gaps impacts and 1% (*n* = 1) experienced 13 impacts. The largest number of survey respondents (*n* = 13, 19%) indicated experiencing 5 workforce gaps impacts. Please see Figure 2 for a representation of how many survey respondents indicated experiencing workforce gaps impacts.

Among those reflecting on the impact of workforce gaps associated with COVID-19 (Table 3), 94% of survey respondents (*n* = 66) indicated experiencing poor operational outcomes from workforce gaps (theme 2.1), and 86% of survey respondents (*n* = 60) had adverse personnel impacts (theme 2.2). 44% of survey respondents (*n* = 31) encountered other workforce gaps impacts. Within these themes, the top operational outcome from gaps was scaling back programs and services (*n* = 54, 77%), and the top adverse personnel impact from gaps was staff burnout (*n* = 59, 84%). The top other workforce gap impact was partnerships with the external organization (*n* = 28, 40%).

### 3.3. Domain 3: Improvement Strategies

The public health workforce is often resourceful, and many focus group participants described ways their agencies could overcome the aforementioned challenges and thus we developed the third domain, improvement strategies. Three themes were developed in this domain: retention strategies (theme 3.1), systems-level changes (theme 3.2), and recruitment/hiring strategies (theme 3.3). Theme 3.1, retention strategies, was defined as approaches taken by focus group participants’ agencies to decrease workforce turnover. The most common retention strategies described by focus group participants included providing employees work flexibility, ensuring supportive leadership, and modifying job positions and duties. Flexibility was often described in the form of work–life balance, such as one focus group participant who said, “If we didn’t offer that [work-from-home options], I think we would have lost people. Flexibility is what allows us to keep a lot of people”. Flexibility was often accompanied by focus group participants discussing the importance of supportive leadership, particularly in light of budgetary challenges, as exemplified by another focus group participant who said,


*…leadership matters when they see that I’m in the trenches with them and we’re making decisions as a team and providing those thank you’s and providing that support and those resources to them. That matters in retention. Because I can’t do anything about the money.*


Modifying employees’ job positions or duties (e.g., job descriptions, hours per week) allowed some agencies to address burnout and increase retention. For example, a focus group participant said, “We really have kind of reorganized and restructured a little bit…and took some of their other normal duties off their plate”.

Systems-level changes, theme 3.2, was defined as modifications focus group participants’ agencies made on a system-wide level resulting in agency improvement. Some of these changes were broad transformational thinking and processes, “we weren’t going to be able to hire unless we got way out of the box”, and others were specific about their strategies such as adding support positions using grants or positions that permit direct billing for services delivered. One focus group participant shared, “we also applied for that workforce grant and we will be employing a strategist”.

Lastly, recruitment/hiring strategies (theme 3.3), was defined as new or innovative methods focus group participants’ agencies used to recruit and hire staff for open positions. These methods often circled back to many of the retention strategies such as providing employees with flexibility as described by one focus group participant, “Flexibility is something we can offer that healthcare cannot”. Other strategies included relaxing position requirements or using equivalencies and recruiting from partner agencies as another focus group participant described, “stealing from your counter[part] counties always happens a lot too”. Though, focus group participants also noted that this last recruitment strategy can worsen existing workforce gaps and exacerbate the aforementioned challenges.

## 4. Discussion

Recruitment and retention of skilled public health professionals nationwide has been a focal point for the public health workforce for over three decades [26,27]. Macroeconomic trends, response to the COVID-19 pandemic, harassment of public health professionals, and the tail-end of Baby Boomer retirements have again brought these issues to the forefront. In 2021, the nationally orientated “Staffing Up” project estimated that state and local health departments needed an 80% increase in the workforce to deliver foundational public health services [28]. The present study explored Minnesota’s local public health practitioners’ perceptions of public health workforce gaps, impacts of those gaps, and lessons learned in light of the COVID-19 pandemic.

Participants discussed workforce gaps and impacts of the gaps that their agencies are experiencing due to the COVID-19 pandemic. Within these gaps and impacts, participants emphasized how being understaffed has limited or altered the services that they provide. One focus group participant noted that local public health agencies struggle with staff recruitment and retention as local health department positions are not necessarily attractive to applicants. Research indicates that this may be a ramification of long-term underfunding of local and state public health agencies [28,29,30], offering less competitive salaries compared to private agencies [31,32], and decreasing retirement benefits in recent years. While there has been significant short-term COVID-19 funding [33], these temporary funds cannot be used to build long-term infrastructure. Many workforce gaps and impacts may have reciprocal effects on each other, though little research has explored this topic that has increased in importance since COVID-19. For example, our participants described experiencing staff burnout, which could impact increased early retirements, forcing the agency to scale back programs and services, leading to a reorganization of programs and priorities. Therefore, we highly recommend that future research explore workforce impacts from COVID-19 and potential cyclical and reciprocal effects.

Despite these hardships, local health officials in Minnesota also discussed lessons learned and ways their agencies can overcome these challenges. First, focus group participants described retention strategies, including workplace flexibility, ensuring supportive leadership, and modifying job positions and duties. Previous literature echoed those strategies and suggested that non-financial incentives, such as housing and improved working conditions, may improve retention and should be sufficiently flexible to target workers’ specific needs [34,35]. Though pay may be one of the most significant drivers of public health employees leaving, dissatisfaction or burnout appear now as significant issues in public health workforce [11,36], leading to higher odds of expressing intent to leave [37]. Furthermore, retention is often contingent on extrinsic employer rewards and individual intrinsic rewards derived from their role and performed work [38]. One primary lesson learned from the COVID-19 pandemic is the importance of public health agencies working to increase workforce retention through non-financial means such as career pathways, employee recognition and satisfaction programs, and employee support to curb some of the impacts of workforce gaps. Additionally, critical for consideration are recruitment and hiring, in which our participants often employed strategies targeting both recruitment and retention. Bringing in young professionals and new public health graduates is imperative to expand the public health workforce [37,39,40]. Fair compensation is essential for recruiting young professionals. Still, numerous non-financial strategies to attract recently graduated students exist, such as increased publication of job opportunities, increased transparency, growth opportunities, collaborative environment, and employer innovation, creativity, and diversity [41]. These strategies may also be effective for recruiting mid-career professionals and are similar to many retention strategies. Since many health departments continue to struggle financially, especially after the COVID-19 pandemic, implementing non-financial recruitment and retention strategies is imperative. Lastly, system-wide changes to the public health workforce have been needed for numerous years [27,42,43], and focus group participants’ discussions of system-wide improvements demonstrate that needs continue. In particular, our participants’ systemic changes focused on retention strategies due to the significant havoc from the COVID-19 pandemic. Therefore, when implementing system-wide workforce changes, practitioners should prioritize retention strategies and demonstrate such prioritization by transparently aligning human and fiscal resources [44].

Though our participants were from Minnesota, the findings may apply to the national public health workforce and likely international settings. Participants’ many workforce gaps and impacts are observed in other areas worldwide [45,46]. Employers should use this study’s findings as a starting point to understand workforce gaps and impacts in their organizations and implement improvement strategies, such as those described above. Additionally, recruitment and retention of early- and mid-career professionals is of utmost importance in responding to and recovering from a global pandemic and building a strong workforce to create a healthy nation.

## 5. Conclusions

Workforce gaps and impacts will likely continue to escalate as the effects on the public health workforce from COVID-19 will be felt for many years to come. Our focus group and survey participants provided many insights into these areas, including local infrastructure gaps, workforce capability and capacity gaps, poor operational outcomes from gaps, adverse personnel changes, and recruitment strategies. These insights described the impacts of the pandemic, including challenges, lessons learned, and opportunities for the public health workforce. Agencies and researchers must explore these and additional impacts and implement quality improvement to create a stronger public health workforce in the future.

## Figures and Tables

**Figure 1 ijerph-19-13084-f001:**
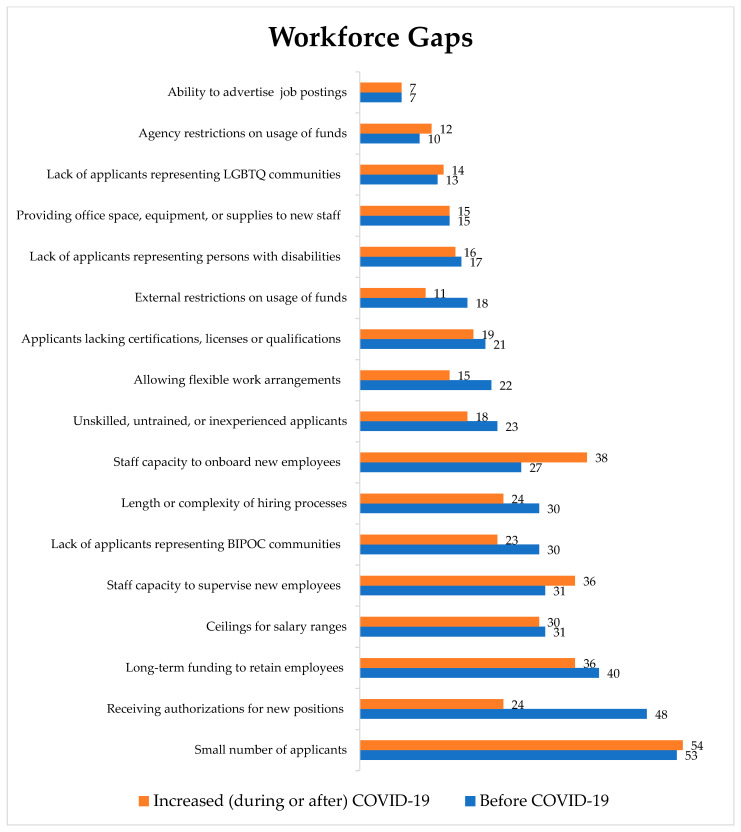
Workforce gaps participant agencies experienced before and increased during or after COVID-19. (*n* = 69 Before COVID-19; *n* = 68 Increased during or after COVID-19). Note: Participants could check more than one workforce gap.

**Figure 2 ijerph-19-13084-f002:**
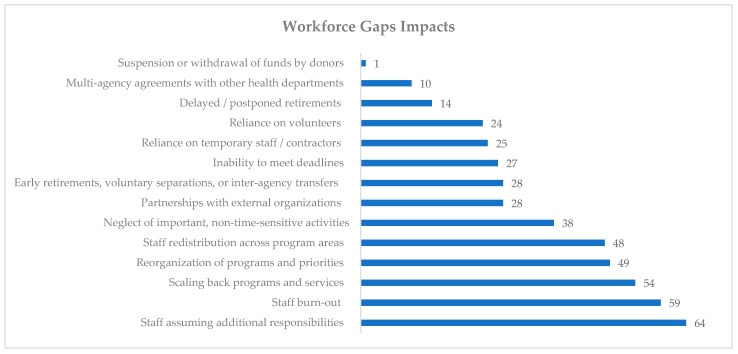
Workforce gaps impacts experienced by survey respondents’ agencies. (*n* = 70). Note: Survey respondents could check more than one impact.

**Table 1 ijerph-19-13084-t001:** Domains, themes, theme definitions, and illustrative quotes. (*n* = 55).

Domain	Theme	Definition	Example Quote
1. Workforce gaps	1.1 Local Infrastructure gaps	Structural or procedural deficiencies that participants identified within their local public health agencies	“do we have the current infrastructure and staffing to support all of that extra and ongoing work? I would say no, not at this time”.
	1.2 Workforce capability and capacity gaps	Needing professionals with appropriate skills (“capability”) and having enough staff to deliver services (“capacity”)	“one of the worries is, is that there won’t be enough people. And we may have to make some decisions about what we can and/or should do as a Health and Human Services Agency and what should the community to do”.
2. Workforce gaps impacts	2.1 Poor operational outcomes due to workforce gaps	Consequences to operational outcomes at agencies due to unfilled staff positions	“One of the things that we have been struggling with for some time is staff, in particular nursing staff, public health nurses who are representative of the community we serve. So that is a huge gap to being able to deliver the most effective service”.
	2.2 Adverse personnel impacts	Negative effects experienced by participants’ agencies due to unfilled staff positions	“It is that stress level again and it’s being understaffed already and having to take on more and more and being asked. I shouldn’t say asked. I let everyone know I know they’re one individual and they can only do so much in one day. But the feeling when you’re passionate about the work that you do of wanting to take on more and more. And yet we don’t have the staff to cover it and so they try and cover it themselves. And then as was indicated just previously, the burnout that happens from that”.
3. Improvement strategies	3.1 Retention strategies	Approaches taken by participants’ agencies to decrease workforce turnover	“assuring that there’s a balance…providing the opportunity to work from home has been a big driver”
	3.2 Systems-level changes	Modifications participants’ agencies made on a system-wide level resulting in agency improvement	“I think the other piece and the recovery period is looking at just sort of re-strengthening the reputation and integrity of public health”.
	3.3 Recruitment/hiring strategies	New or innovative methods participants’ agencies used to recruit and hire staff for open positions	“We’ve been talking about this far, probably a little bit pre-COVID. We had started having these conversations and moved it little bits here and there, and just getting HR [human resources] to allow a little bit more flexibility and recognize that. I would say we’re still in the process of updating formally job descriptions and what’s listed as minimum quals. But having those substitutions and things”.

**Table 2 ijerph-19-13084-t002:** Workforce gaps survey respondents’ agencies experienced before and increased during or after COVID-19 sorted by focus group themes. (*n* = 69 Before COVID-19; *n* = 68 Increased during or after COVID-19).

Workforce Gaps	Before COVID-19		Increased (during or after) COVID-19	
	*n*	%	*n*	%
Local Infrastructure Gaps	65	94%	55	81%
Receiving authorizations for new positions	48	70%	24	35%
Long-term funding to retain employees	40	58%	36	53%
Ceilings for salary ranges	31	45%	30	44%
Length or complexity of hiring processes	30	43%	24	35%
Allowing flexible work arrangements	22	32%	15	22%
External restrictions on usage of funds	18	26%	11	16%
Providing office space, equipment, or supplies to new staff	15	22%	15	22%
Agency restrictions on usage of funds	10	14%	12	18%
Workforce Capability and Capacity Gaps	64	93%	64	94%
Small number of applicants	53	77%	54	79%
Staff capacity to supervise new employees	31	45%	36	53%
Lack of applicants representing BIPOC communities	30	43%	23	34%
Staff capacity to onboard new employees	27	39%	38	56%
Unskilled, untrained, or inexperienced applicants	23	33%	18	26%
Applicants lacking certifications, licenses or qualifications	21	30%	19	28%
Lack of applicants representing persons with disabilities	17	25%	16	24%
Lack of applicants representing LGBTQ communities	13	19%	14	21%
Ability to advertise job postings	7	10%	7	10%

Note: Survey respondents could check more than one workforce gap.

**Table 3 ijerph-19-13084-t003:** Workforce gaps impacts experienced by survey respondents’ agencies sorted by focus group themes. (*n* = 70).

Workforce Gaps Impacts	*n*	%
Poor Operational Outcomes from Workforce Gaps	66	94%
Scaling back programs and services	54	77%
Reorganization of programs and priorities	49	70%
Neglect of important, non-time-sensitive activities	38	54%
Inability to meet deadlines	27	39%
Reliance on temporary staff/contractors	25	36%
Reliance on volunteers	24	34%
Suspension or withdrawal of funds by donors	1	1%
Staff assuming additional responsibilities	64	91%
Staff redistribution across program areas	48	69%
Adverse Personnel Impacts	60	86%
Staff burnout	59	84%
Early retirements, voluntary separations, or inter-agency transfers	28	40%
Delayed/postponed retirements	14	20%
Other	31	44%
Partnerships with external organizations	28	40%
Multi-agency agreements with other health departments	10	14%

Note: Survey respondents could check more than one impact.

## Data Availability

The data are not available due to concerns around data confidentiality of key informants and survey respondents.

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
