# Peer review of "Public Health Workforce Gaps, Impacts, and Improvement Strategies from COVID-19"

_ijerph, 2022, doi:10.3390/ijerph192013084_

Round 1
Reviewer 1 Report
This is an interesting work integrating the qualitative and quantitative investigations on the Public health workforce problems under the impact of COVID-19. I enjoy reading this manuscript. Here are my comments.
Can you please give more introductions on the public health workforce at different level of governments in the introduction? Particularly, the different duties and responsibilities during the COVID-19 if possible.
Which criteria is used to select the focus group participants? Is it possibly relevant to the COVID-19 severity?
In section 2.2, it seems that the questions could be answered by investigation. Is there any other data support available in addition to the questionnaire.
The quality of figures are poor.
The negative impact of COVID-19 can result in many socioeconomic burdens. But the asymptomatic infections should also be considered. Please see this article https://doi.org/10.3390/fractalfract6040197
Will your study offer some guidance or comparison with other states or countries.
Line 87-88 QUAL->qual-> integration. Why capitalized QUAL only?
Author Response
Thank you for your notes. Please see the table below for our responses.
Feedback |
Response |
Reviewer 1 |
|
This is an interesting work integrating the qualitative and quantitative investigations on the Public health workforce problems under the impact of COVID-19. I enjoy reading this manuscript. Here are my comments. |
Thank you for your comments. Please see our responses below. |
Can you please give more introductions on the public health workforce at different level of governments in the introduction? Particularly, the different duties and responsibilities during the COVID-19 if possible. |
This manuscript focuses on local public health rather than all levels. Therefore, we added in a clarification earlier in the manuscript noting that we are focused on local public health (line 63) and added a sentence describing their primary responsibilities during COVID-19 (lines 35-38). |
Which criteria is used to select the focus group participants? Is it possibly relevant to the COVID-19 severity? |
We added a sentence further clarifying participant selection in lines 103 – 105). |
In section 2.2, it seems that the questions could be answered by investigation. Is there any other data support available in addition to the questionnaire. |
Thank you for the comment. The study design leverages a mixed methods approach where the investigation combines qualitative inquiry via focus groups with a quantitative survey methodology. |
The quality of figures are poor. |
These figures have been uploaded as separate files, which are clearer. |
The negative impact of COVID-19 can result in many socioeconomic burdens. But the asymptomatic infections should also be considered. Please see this article https://doi.org/10.3390/fractalfract6040197 |
Thank you for the reference Pan et al. (2022). The focus of this research article is focus group feedback elicited by participants with respect to workforce gaps and impacts with respect to COVID-19, both symptomatic and asymptomatic. The suggested paper is valuable, but does not bear on the issues in the present research. |
Will your study offer some guidance or comparison with other states or countries. |
Thank you for the suggestion. While we do not directly name other states or countries, we do allude to others throughout the discussion. Please see lines 349 – 350, 390-392, 394-396. |
Line 87-88 QUAL->qual-> integration. Why capitalized QUAL only? |
Per Creswell (Creswell, J.W.; Plano Clark, V.L. Designing and conducting mixed methods research, 3rd ed.; Sage Publications Inc.: 2018), “Qual” is capitalized and not “quant” as capitalization indicates the method priority. Please see lines 93-97 for this explanation. |
Reviewer 2 Report
The methods and analysis presented in this manuscript are sound and add nuance to the national discussion of urgent needs in the public health sector. The manuscript would benefit from a thorough copyedit and reformatting of the tables. Repetition of the theme numbers is not necessary. A few specific comments follow with line references.
45: not just considering--they are actually leaving public health
58-59: this is an example among several of awkward syntax
75-76: a citation would be helpful here and why the reference to the social service safety net?
81: This sentence reads as if it were lifted from the grant proposal (future tense.
266: 19% is not "most"
281 and following: review text to make sure the distinctions between data in Table 6 and Fig. 2 are clearer to the reader
284-6: this is an example of sentences that seem to be missing words, again calling for a careful read and copyedit
296: correct term is "exemplified"
318: should this be "counterpart counties" rather than "counter counties"?
397: the word "potentially" weakens the concluding sentence
Author Response
Thank you for your notes. Please see the table below for our responses.
Feedback |
Response |
Reviewer 2 |
|
The methods and analysis presented in this manuscript are sound and add nuance to the national discussion of urgent needs in the public health sector. |
Thank you for your comments. Please see our responses below. |
The manuscript would benefit from a thorough copyedit and reformatting of the tables |
We have edited the paper and will work with editorial to ensure tables are formatted correctly. |
Repetition of the theme numbers is not necessary. |
Thank you for the suggestion and we will defer to editorial as to whether to include theme numbers or not. |
A few specific comments follow with line references. |
|
45: not just considering--they are actually leaving public health |
This reference uses the words “planning to leave,” thus we change the wording to match. |
58-59: this is an example among several of awkward syntax |
These awkward syntaxes have been reviewed and revised through the thorough copyedit. |
75-76: a citation would be helpful here and why the reference to the social service safety net? |
We added a citation for this statement. We also added “social service safety nets” to our opening paragraph, which ties in with this statement. |
81: This sentence reads as if it were lifted from the grant proposal (future tense. |
This sentence has been revised to keep the tenses aligned. |
266: 19% is not "most" |
This sentence has been revised for clarity . |
281 and following: review text to make sure the distinctions between data in Table 6 and Fig. 2 are clearer to the reader |
We only have three tables, so we are unsure which table is being referred to. However, during our thorough copyedit, we paid close attention to the text describing the tables and figures to ensure the reader can distinguish between the data presented in each. |
284-6: this is an example of sentences that seem to be missing words, again calling for a careful read and copyedit |
We have completed a careful read and copyedit, which included careful attention to phrasing. |
296: correct term is "exemplified" |
This word has been changed. |
318: should this be "counterpart counties" rather than "counter counties"? |
Since this is in a participant’s quote, it should be left as is. To add clarity, we added “[part]” after “counter.” |
397: the word "potentially" weakens the concluding sentence |
The word “potentially” has been removed. |
Round 2
Reviewer 1 Report
The figures are still of low quality. I can hardly recognize the legends and labels.
Author Response
Thank you for your review. We have re-inserted the figures and increased the font size to improve quality. We also have uploaded the figures separately so the editors can import them into the final manuscript without losing resolution.